# The Need for Cybersecurity in Industrial Revolution and Smart Cities

**DOI:** 10.3390/s23010120

**Published:** 2022-12-23

**Authors:** Antonio Clim, Andrei Toma, Răzvan Daniel Zota, Radu Constantinescu

**Affiliations:** The Department of Economic Informatics and Cybernetics, The Bucharest University of Economic Studies, 010552 Bucharest, Romania

**Keywords:** smart city, cybersecurity, industry 4.0, industry 5.0, Internet of Things, smart factory, smart mobility

## Abstract

Cities have grown in development and sophistication throughout human history. Smart cities are the current incarnation of this process, with increased complexity and social importance. This complexity has come to involve significant digital components and has thus come to raise the associated cybersecurity concerns. Major security relevant events can cascade into the connected systems making up a smart city, causing significant disruption of function and economic damage. The present paper aims to survey the landscape of scientific publication related to cybersecurity-related issues in relation to smart cities. Relevant papers were selected based on the number of citations and the quality of the publishing journal as a proxy indicator for scientific relevance. Cybersecurity will be shown to be reflected in the selected literature as an extremely relevant concern in the operation of smart cities. Generally, cybersecurity is implemented in actual cities through the concerted application of both mature existing technologies and emerging new approaches.

## 1. Introduction

Generally, a ‘smart city’ is one that integrates sophisticated technology to automate service provision [1]. This is, however, largely limited to varying extents by the resource availability of the city, level of available technology, and acceptance of the city dwellers of such change. There is hence no global standard definition of a smart city. It is, rather, a concept that defines the integration of existing industrial processes with advanced technology for better data collection and communication to achieve efficiency.

Different and often complementary points of view on a smart city include the efficient, green, technologically advanced, sustainable, and socially inclusive city [2]. However, smart cities can differ greatly from each other, due to variable characteristics, such as economic factors, infrastructure, or governance. Broadly speaking, the main elements of a smart city include: infrastructure and other utilities, traffic management and transportation, buildings and advanced technology used for communication within a city and between cities [3]. In this sense, in order to be able to integrate all the components of a smart city and centralize the control of its resources, it is necessary to develop a centralized system that uses advanced technologies related to artificial intelligence, cloud computing or IoT (Internet of Things) [4]. Such a complex IT system, integrated at the level of a city, involves a series of physical and electronic components that must interact in a network to achieve the desired results.

Smart cities therefore possess complex, large and interdependent IT systems, which exposes them to several technical, security and privacy challenges [5]. Our study emphasizes the aspects related to security, closely related to confidentiality and data integrity. Security in the technical scenario involves cyber security and involves the protection of the entire network, its data and information against malicious activities, as they incorporate various technologies, such as portable devices—smart phones and radio frequency identification systems, for example [5].

The efficiency of a smart city is indicated by the computing capacity of the IT system as well as the high level of integration of its various components. Such a computer system depends on these computing components and can fail if any component of the system is compromised by a cyber-attack or serious error, such as not performing security tests before installing it in the system. Whether simple or critical, any system component is important to the overall integrity and proper operation and longevity of the system.

As seen in Figure 1, each stage of industrial revolution has increased the efficiency of industrial processes, while raising the complexity of the industrial systems themselves [6,7]. The evolution of Information Communication Technology, and its integration in production processes, has also changed conventional industrial processes and improved organizational management [8].

These developments have not been without controversy, however. Some examples of developments stirring global debate are self-driving cars or increased supermarket automation. The development of these features has been catalyzed by advances in artificial intelligence as well as systems based on the collaboration of human and machine as resources to form a network that executes commands with an ability that is close to human intelligence [10,11]. With the incorporation of collaborative robots (cobots) into the manufacturing and production processes, simulation is now extensively used in operations to exploit real-time information in order to improve quality through the construction of virtual models [12,13,14]. These simulations can be made in 3D or 2D and can be used to monitor processes such as cycle times and energy consumption, which help reduce failure and wastage along the production line [6,15]. The evolution of these industries, therefore, has had the impact of transforming ordinary machines into intelligent automated machines that can improve the operating and management performance of the network in which they interact. Sectors which have more quickly integrated these trends have increased significantly in added value [16]; however, this increase has been accompanied by increased costs related to data security. Thus, data security, and in a larger sense cybersecurity, has to be incorporated as a primary concern in these evolving networks and systems.

With increased adoption of approaches characteristic of Industries 4.0 and 5.0, efficiency gains have come alongside new operational challenges [17]. Industry 4.0 and 5.0 systems have complex structures, made up of various interdependent components. If any of these components is compromised, this can pose a severe security threat to the system, escalating into interruptions in the system’s functioning. The integration of these components makes the system dangerously susceptible to attacks, as when a single component is compromised, the entire chain may fail. If the system was not programmed to mitigate such attacks, their impact could cause a halt of all operations carried out by the network [18,19]. As such, cybersecurity represents a central consideration and is paramount to their success. The issue of cybersecurity in this environment must be addressed starting from the design stage, and security measures must be implemented in existing systems before fully integrating them with the network.

Given these new security concerns, a study evaluating cybersecurity as applied to industry 4.0 and 5.0 systems in terms of both existing capabilities and possible future improvements should prove valuable in terms of understanding both the present situation as well as upcoming challenges.

## 2. Methodology

This review of the literature adopted specific procedures and approaches that were relevant for systematic research. Therefore, its main objective was to shed light on the topic of discussion by recording, identifying, understanding, and sharing information, as well as making sense of the topic of interest. Therefore, when writing this review of the literature, it was essential to consider its significance in general research, because such rigorous knowledge shared in this section is indispensable to keep up with the ever-growing cybersecurity literature that is vital in helping practitioners and relevant personnel to come up with findings and then assess and synthesize the issues discussed in conceptual and empirical discussions. As a result, in this literature review, systematic research was conducted, involving an extensive collection of various published works by different authors regarding the topic of discussion and assessed topics that could be learned through considerable discussion.

Moreover, the methodology of this discussion took two primary forms, focusing on the existing literature and on the gaps in this research topic that were addressed in these empirical studies. This way, it was possible to develop a theoretical foundation for the topic of discussion, demonstrate the existence of the research, and justify the research’s contribution towards the cumulative knowledge through validation of the approaches discussed in the literature review. Google scholar has been used for searches, using specific key-words, and groups thereof, as shown in Table 1. The first run of the search (February 2022, 2nd version of this study), using main terms from Table 1 and an arbitrary interval from 2012, returned 16,100 results. Eliminating old versions of the same studies, the number of valid results decreased to 4,578 titles (showing an average of 3.5 versions per title). After filtering the articles which appeared in more than one repository and articles in other languages than English, there remained 1,033 published studies, from which were picked those open sources (mainly). On this group, additional terms were applied, resulting in a base of 647 items to be directly analyzed and classified. Reading the titles, key-words and abstracts and eliminating results with up to 10 citations, and also choosing the titles in journals classified as SCI (Science Citation Index), SCIE (Science Citation Index Expanded) and/or ESCI (Emerging Sources Citation Index), the list was reduced at 192. The resulting proportions between articles vs. conferences papers vs. books (or chapter in a book) were 78% vs. 14% vs. 8%. After a qualitative sorting step, only 102 scientific research studies were chosen and cited.

In this research, not only have scientific articles been used and cited (Table 2) but also industry-related reports from companies (Verizon, Microsoft, PwC, Deloitte, KPMG, Ernst&Young, etc.), agencies reports (ENISA, ECSO, CEPS, EU Parliament, FAO-UN, etc.), standards (from ISO), etc., as shown in Table 3.

The chosen studies are detailed in Table 3, by main editors and number of citations (up to 15 of November 2022).

Additionally, the second form of methodology used for this discussion entails the valuable and original work of discussion, which provides a foundation for the work and gives a concrete perspective for other studies. Therefore, having a review discussion will offer an all-encompassing purpose of developing a synthesis of research pertinent to this topic without the collection or evaluation of any primary data.

## 3. Correlation between Industry 4.0 and 5.0 with ICT and Cybersecurity

Globally, the manufacturing industry has been adopting ICT over the years in an attempt to facilitate its activities, and as a result, this adoption of technology has offered opportunities for disruptive strategies when it comes to production, development, logistics and the value chain [20]. However, this adoption of ICT has blurred the boundaries between the virtual and the real world, leading to cyber-physical production systems (CPPSs) [21]. This is in fact what led to the emergence of Industry 4.0 (see Figure 2) and Industry 5.0. Nevertheless, it is worth mentioning that implementation of ICT is associated with other operational risk factors associated with smart and connected manufacturers, as well as networks for the digital supply networks [22]. The interconnection of this network with industry-driven operations and the speed with which digital technology transformed the industry means increased vulnerability to attacks that have more substantial effects than it has ever happened before [23] from a cyber-security perspective. As a result, these events have resulted in numerous studies that seek to review some of the critical aspects of the industry in relation to technological developments, such as the defining characteristics of cybersecurity, as it has been one of the main concerns in which information networks have been adopted for different operational activities [24,25].

## 4. Digitalization of Industry 4.0 and Industry 5.0

Industry 4.0 is based on autonomous devices and other technologies that have facilitated the production process throughout the value chain. Therefore, this model of a smart factory based on these technological advances includes a section whereby the physical processes of the factory are monitored by systems controlled by computers [27]. Such systems have been developed in such a way that they contain a virtual copy of the physical world, producing decentralized decisions based on mechanisms surrounding self-organizations [28]. Consequently, this concept will lead to more digitization of the manufacturing industry so that physical objects will be effortlessly integrated into the information technology’s network. Thus, this will enable manufacturing systems to have vertical networking with their relevant business processes within an organization, as well as for the horizontally connected systems to have spatial dispersion value networks which can be regulated in real-time, from the moment an order has been placed up until the outgoing logistics of that particular order. As a result, we have differences across industries and the services offered tend to be less pertinent, since the digital technologies have a connection with the industrial products and services into mixed products, which are considered to be the hybrids, in that they do not only contain goods or services [28,29]. Therefore, it is worth mentioning how the Internet of Things, as well as the Internet of Services, are deemed the main elements within Industry 4.0 (see Figure 3).

Alternatively, another perspective on Industry 4.0 is with regard to CPPSs, which involve an online network with service machines structured similarly to social networks [30]. Therefore, these networks will connect IT with mechanical and electrical components and will communicate with one another through a network. For instance, advancements such as radio frequency identification (RFID) technology have been among the earliest forms of this technology, which has been under use since its inception. In the contemporary setting, smart machines will gradually share information regarding current levels of stock, faults, and issues, as well as changes in orders or demand levels [31,32,33]. Therefore, deadlines and processes are under extensive coordination in order to enhance efficiency, as well as improve capacity application, intime, as well as the quality of production, development, purchasing, and marketing. Therefore, CPPSs will not just network the machines within an industry on a daily basis, they will also develop a smart network surrounding ICT systems, machines, properties, smart products, and stakeholders throughout the life cycle of a product and the value chain [34,35,36,37]. Moreover, another significant aspect of Industry 4.0 is the possibility of interacting with other smart infrastructures, such as those associated with smart homes and buildings, smart grid, smart logistics, and smart mobility, as presented in Figure 4.

The connection to both business and social networks through the business web as well as the social web will play a mandatory role in the digital transformation of Industry 4.0. Furthermore, these industry-based interfaces and networks will be integrated into the IoT, such as data, people, and services [39,40]. Therefore, some of these smart industries will depend on various contemporary innovative technological advancements. For example, the application of ICT in the digitization of information and the integration of the system throughout different stages of the development of a product and its appropriate use, such as logistics and supply, is to be made within the company or its limits [41]. Second, network communications entail Internet and wireless technologies that allow people, products, and systems to interact, as well as connecting machines in manufacturing industries with relevant suppliers and distributors [31]. Third, cyber-physical systems which employ ICT in monitoring, as well as regulating physical systems and processes entail intelligent robots and embedded sensors, since they can be configured to adapt to their immediate product. In other instances, additive manufacturing devices, such as 3D printing devices, are used [42]. Moreover, the collection of different quantities of information and their exploitation and evaluation within the factory or by having cloud computing and big data analysis, virtualization, simulation, and modeling within the design of a given product, as well as the development of the manufacturing process, and the immense ICT-based support for the human worker, involve augmented reality, robots and cobots, and other intelligent tools [43,44]. These tools are represented in Figure 5.

## 5. Major Vulnerabilities of Digital Technologies in Industries

The nature of cyberattacks directed at contemporary industries is based on targeting of well-known vulnerabilities within digital technologies which are central to Industry 4.0 and Industry 5.0 [46]. The first source of vulnerabilities is the continued use of insecure legacy systems that have reduced maintenance. This is common for cities which have been stacking layers of old infrastructure, which depends on old technology and software. Moreover, these systems have not been updated or upgraded in a very long time and they have not evolved into more advanced and secure systems [47]. Therefore, having such technologies for industry can give rise to vulnerabilities when it comes to contemporary systems, since they are likely to contain forever-day exploits that are understood to be the loopholes in legacy software products that are no longer supported by their vendors, meaning that the system will never be patched [48]. Additionally, even when considering new technologies, it will be hard to determine and roll out the patches on the critical operational systems, which will always need to be on.

Another major vulnerability is the use of weakly tested software, as well as weak data encryption protocols to secure the systems used in these industries. Studies have stated that there are about 30 errors and other exploit bugs in about 1000 lines of code [49,50], and when it comes to a contemporary heterogeneous system which is being used in industries and smart cities, it has millions of lines of code that have the potential of producing zero-day exploits for various system hacks, malware, and viruses. In addition, studies of industries and smart cities have shown that most of their IT systems were developed without maximum security. Here we refer to the fact that there are no user authentication methods and default or weak passwords are used [51]. Furthermore, vendors, stakeholders, and governments within these industries have used their technological systems without conducting in-depth cybersecurity testing and drills, which would have been necessary considering some of the security issues which can arise based on how different systems are operated.

Thirdly, vulnerabilities can produce cascade effects, which can cause cyberattacks, and the factors involve the interrelations between industrial technologies and the system. This creates the grounds to develop various cascade effects through disruptions or failures, hence generating knock-on effects which cause more failure in different significant services, such as utilities. For example, when a telecommunications system is under a cyber-attack, the infrastructure is overloaded, and this may lead to failures of emergency response or traffic management system. Therefore, this makes the issue a significant security risk within an urban operating system, whereby there are various systems that have been interconnected, reflecting a system-of-systems approach in the management of services within the city and its infrastructures. As a result, it will nullify the mitigating impact of using the siloed approach, which is perceived as an entirely individual system, whereby there is a physically independent telecommunication cabling as well as a source of power among other systems [52,53,54]. Therefore, in case there is a successful cyber-attack on the electricity grid in a specific place, there will be sizeable cascading effects because that service supports numerous activities such as distributing power to workplaces, homes, hospitals, and many other infrastructures.

Lastly, human error and intentional malfeasance of dissatisfied personnel within a workplace can be another reason for system vulnerabilities because technical exploits can be immensely assisted by the errors people make. For example, when an employee responds to phishing messages or emails, the insertion of infected external drives into computers and installation of malware and viruses can be some of the ways these actors can play a crucial role towards a cyber-attack [55,56]. There are other instances whereby the software of a particular infrastructure could be weak in their system designs, in the sense that they can comfortably and secretively be sabotaged with the help of the rogue employees or former employees (see Figure 6). Therefore, it is worth noting that the aforementioned vulnerabilities can be worsened by various factors, considering that at times it is unclear who is mandated to maintain the security across the complex infrastructure and system. Moreover, vulnerabilities also appear when there are many stakeholders and other institutions which have worked together in the development of designs, supplying software and hardware, as well as in the operation and use of different components.

## 6. Cybersecurity Risks in Industry 4.0 and 5.0

The technological advancements that constitute the basis for Industry 4.0 have raised various issues concerning their security and overall safety, therefore stating that these developments in the technological paradigm are a direct indication of opportunities and challenges [58]. According to the study conducted for the Associated Chambers of Commerce of India (ASSOCHAM) [59], Industry 4.0 and Industry 5.0 will prioritize talent over money as a form of investment because the use of elements such as IoT, Artificial Intelligence, blockchain, and other emerging technologies is growing exponentially. These elements can be used to improve governance, consulting, and general digitalization of the major industries [60]. The adoption and implementation of these technologies will improve various operational capacities, talent acquisition, and the delivery of applications in various organizations and sectors. Therefore, the process of integrating a concept within an industry is actually about opening up the IT infrastructure of that industry, thus making it more vulnerable to errors and susceptible to cyber-attacks.

Moreover, it is worth noting that cyber intruders will not stop at anything, considering that as the industry advances, they will also refine their techniques for breaching systems [61]. As a result, such cyber-attacks targeting penetration into the control systems of various organizations will become a threat to the facilities’ production. In case of such attacks, industry computers are usually controlled from a remote location, and in turn, the intruders can have access or spy on their targeted objective [62,63]. Such attacks have been made possible by malware that exploits firewalls or security holes, and oftentimes, network monitoring software cannot detect such intrusions.

Considering that these technological opportunities are also in tandem with their associated cybersecurity risks, it would be necessary for industries to have a complete understanding of the impact of adopting these technologies on their security [64,65,66]. For such industries to capitalize on the fourth and fifth industrial revolutions, they will need to have the necessary contingencies and measures in place that will ensure that their cybersecurity is in tandem with Industry 4.0 and Industry 5.0 [67,68]. Therefore, factors such as integrated data protection frameworks, as well as sector-specific security baselines will help industries find sustainable advantages from their contemporary technological revolution. It is worth noting that the government, as well as other institutions, are also faced with challenges in their cybersecurity, as they are required to have their platforms resilient to threats so that they can facilitate the adoption of ever-changing and disrupting technology [56,69]. In addition, conventional information security practices could offer the required strategies, but this would not offer sufficient protection for industries considering how technology has been evolving over the years.

Regarding the promise of the impact of cybersecurity plans on organizations, some of the appropriate frameworks that industries can focus on as well as commit to should have some common attributes. For example, the framework should be able to develop capabilities that can detect threats and respond to them adequately and proactively, as well as be able to make use of artificial intelligence in identifying patterns so that IT systems can be monitored through smart technology [70,71,72,73]. Such frameworks should also offer an integrated approach towards cybersecurity, whereby there will be a holistic strategy towards the landscape of a threat instead of implementing security technologies in silos. Lastly, it is essential for the framework to create a well-built connection with its respective institutions throughout its sectors and those of the government for the purposes of sharing research, intelligence, information, and capacity development. As these technologies are developing and scaling up, it is essential to involve human resources, training, and education [74,75,76] as the cornerstones of industrial progress, because upon the successful adoption of these technologies, there will be demand for new skills and expertise.

## 7. Security Scenarios

Developing and managing a smart city or an industrial revolution as a whole using a computerized network can be challenging in terms of ensuring security for the components and various elements due to the interconnectivity of these elements and the dependence of the whole network on its components. These security risks can be discussed as security of the system infrastructure (SSI).

The physical components of a city’s smart network can be faced with various accidental or intentional attacks, which can cause various degrees of damage depending on the preparedness for such attacks. Below are some of the security issues that can affect city infrastructure, including electricity, water supply, sewer lines and buildings along streets.

Theft: this affects city infrastructure when physical and intangible materials such as data, software and credentials are stolen and put to harmful use.DoS attacks may be used to impede the availability of services. Denial of Service may overflood connections until devices connected to it are blocked or the connection shut down.Spying on the installed communication channels may be dangerous if an attacker is able to extract sensitive information.

Other threats may be related to environmental impact, software crashes, and/or hardware failure in such a way that they affect the integrity of the system and cause inefficiency along the production lines. These threat cases can be addressed through the authentication that is provided by the control packet in MQTT (Message Queuing Telemetry Transport) [77,78]. MQTT clients can also be provided with different items from the server to perform authentication in addition to passwords and usernames [77,79].

Critical systems that manage the transportation sector—for example, air transport—are of particular interest as they can potentially cause catastrophes if a hack were to happen while a plane was airborne. They may also cause considerable damage and loss of time (traffic jam or accidents) if the systems controlling land transport media were to be tampered with.

Building management systems that are poorly designed or that do not include proper security features expose the inhabitants of such buildings to insecurity, especially if such buildings do not incorporate appropriate reporting systems or alternative safety measures.

Communication networks in a smart city incorporate cyber-physical objects that are interconnected through advanced technologies such as WIFI or GSM. These objects need security measures to facilitate their continued operation within the network and to ensure that they work as a unit [80,81]. Moreover, these security concerns need to be addressed during the deployment of these objects.

A smart city that has central traffic control uses cameras for control and data collection. These also need to be protected using encryption methods and password protection to ensure the privacy and security of the data collected.

## 8. Cybersecurity for Smart Cities

When it comes to smart cities, cybersecurity is a contentious topic, and the primary purpose of the development of such cities is to optimize cities around the world in a dynamic way so that the quality of life of people can be improved through the application of ICT. These smart cities differ when it comes to the range of implemented smart technologies, considering that the evolution of connected cities is prevalent in data exchange in a large-scale context [82]. Such cities are susceptible to cyberattacks of different forms, which are meant to disrupt, deceive, destroy, alter, or degrade the computer and ICT systems, such as networks and programs which have been set in place to run the city. Therefore, cyberattacks manifest themselves in three different forms, with a focus on operational systems [83]. Whereas availability attacks focus on closing the system down, thus impeding the use of services, integrity attacks are supposed to enter a system to affect the state of information present in that system through viruses and malware, without causing alarm to the rightful operator. Third, confidentiality attacks are meant to retrieve information and keep tabs on various activities [55]. Such attacks towards the industry can be conducted by different players such as hackers, government agencies, criminals, and terror groups, among many others. Therefore, it should be noted that the growth of ICT and its wide application in different industrial sectors has resulted in an exponential increase in the number of perpetrators of cyberattacks with different intentions.

The complex nature of smart cities, which involve significantly higher levels of data exchange (as represented in Figure 7), will lead to automated control of various services [84,85,86] and assets, thus leading to a high level of automation and interconnection between different services offered by the city. 

This will increase the demand for cybersecurity, with the goal of protecting vital ICT aspects of the city, such as privacy, data exchange, and the safety of people within these cities [75,88,89]. However, it is essential to note that currently there are no coordinated standards and guidelines to model such data exchanges that occur throughout the city. As a result, this has forced some of the IPT (intelligent public transport) municipalities, operators, legislators, manufacturers, vendors, and solution providers to adopt particular solutions that have low scalability and cater to different needs [90]. In the current context, it is difficult for IPT operators to have a cybersecurity policy that can be applied in the process of defining significant assets. Furthermore, cybersecurity knowledge and spending in the context of IPT seem to be low, as there are few measures in place and many cybersecurity contingencies.

Since the early days of smart cities, there has been a growing concern for identifying the dimensions that can be used for their characterization, but there is no consensus so far on the factors to consider in the development of more intelligent and sustainable cities [91]. Therefore, it is imperative to understand the concept of smart cities from the ICT context, with the purpose of discussing some of the technologies that make this form of advancement possible, as well as reflecting on its vulnerabilities, threats, along with how to mitigate and ensure that smart cities are safe and conducive to digitalization [92]. For example, the application of big data from satellite imagery that include sensor networks is crucial for improving the measurement of both developmental and environmental indicators. Such research can be possible given that there are about 6 billion smartphones in the world, with 80% of these phones in developing countries, which provides sufficient coverage to give both researchers and prospective investors the social sensors to retrieve the necessary information [93]. Therefore, big data will change how people live and interact throughout their daily activities, and this will be influenced by the spread of mobile network and cloud computing, and as a result, it will offer farther-reaching effects within the ICT sector [94,95]. Therefore, the ICT sector is projected to become a more considerable investment for smart cities in the coming years.

The concept of smart cities is rapidly growing and, as a result, smart cities are spreading throughout the world (see Figure 8). 

Governments have facilitated this spread by implementing relevant guidelines and policies, offering a better platform for the establishment of these cities [97]. Right now, more than 55% of the world’s population live in urban areas [98]. Additionally, there are around 150 smart cities in the world, depending on the classification criteria [99,100]. The application of innovative ICT is critical to enhancing different programs and initiatives associated with smart technology and can provide the premises to develop some of the most advanced smart cities in the world that integrate sustainability in different forms, such as green cities and those with low carbon emissions [101,102]. It is worth considering that cities have dense populations and, hence, have significant development in industries and commerce and other functional zones, such as the commercial, industrial, administrative, and residential sectors. Therefore, for smart cities, they will use ICT and other digital technologies to facilitate services, sustainability, and harmonious living through the smart application of different coordinated natural and social developments [103,104,105,106]. Furthermore, some researchers point out that different regions have distinct attributes that make each smart city different, according to their natural environment, resource investment, social attributes, and the overall objective [83,106,107,108], but there are common indicators that determine how these cities can employ different strategies to ensure effectiveness and profitability.

It is equally important to note that there are diverse measures that may not widely apply within the cybersecurity standards that align with the demand of the IPT and with the widely employed good practices [109,110]. Therefore, for the development of cybersecurity guidelines, it is vital to consider the high-level architecture model in order to better understand the critical sectors that can protect the industries from cyberattacks. With the knowledge that industries and smart cities have various levels of maturity, the architecture model will have to focus on the association in these smart cities and industries from the point of the IPT operators, thus integrating the functional process and data exchange between different stakeholders [111]. Some of the particular threats associated with data exchange between IPT operators, as well as the relevant stakeholders in relation to their possible consequences, will vary based on the maturity of the smart city [112,113,114]. As a result, this will reveal the multifaceted nature of threats, as well as their dangers towards information and data, organizational structure, the applications, and their related technologies, culminating with the whole framework pertinent for IPT.

## 9. Characterizing and Quantifying Malice

Through research, human beings have been categorized as the arrestees of malice in computer-based networks and systems. Quantifying these together with the economic motivation to carry out such activity and the knowledge of the executioner is vital to approximating malice [115]. Although very little research has been conducted in this area of human interaction and attributes, malice has been classified into micro, meso, and macro levels, and all aspects of this category are a potential cause of cyber insecurity [57]. Therefore, it follows that an individual can act maliciously due to their personality or due to their interactions with others. According to this characterization, malice is viewed as sociotechnical and would be useful in determining optimal levels that can be adopted in the process of integrating human factors when evaluating threats to cybersecurity [116]. Human thoughts and behavior are vital aspects that determine that they indulge in malicious acts, but it is almost impossible to categorize such acts. It is even harder to come up with an index that can be used to standardize the behavior of human beings, since everyone has a divergent approach and ideas on how to deal with every situation. Not even a standard psychological test can be formulated to determine this behavior [117].

However, an individual’s personality can be related to their cultural attributes, and since these can influence their behavior, they can also influence the malicious nature [118,119].

## 10. Future Directions in Cybersecurity

The cyber domain is a multifaceted sector, considering how it has amalgamated different professions and is relevant in almost every industry, since it involves various aspects such as the process of connecting online devices and creating a platform that gives people the opportunity to interact with these devices, thus revealing how these devices have influenced different components of their lives [120]. Therefore, it is worth mentioning that the cyber paradigm is crucial, as it has managed to influence almost every aspect of contemporary life, such as healthcare, powering homes, transportation, and multiple other actions that people perform in their daily lives [121]. As time passes, the number of connected devices and their usage will increase, and as a result of this, the convolution of cyberinfrastructure will exponentially increase. However, as these numbers increase, another major problem associated with the issue is that more vulnerable devices will emerge from this phenomenon, hence the need to have a cybersecurity workforce that will support the development of cyber infrastructure and thus protect networks [122,123]. Consequently, with the inception of cybersecurity and the fact that this issue is critical in maintaining the integrity of cyberspace for many industries, there has been a growing need and demand for the development of various cybersecurity workforces and the pertinent framework that can offer some of the crucial roles of the cyber workforce.

## 11. Conclusions

Cybersecurity is a vital component in the implementation of smart cities, and it is an important consideration for various data technologies and infrastructures. However, the integration and interdependence of the various components of the management systems influence the cybersecurity.

In an era where Industry 4.0 and 5.0 define many aspects that people interact with, such as IoT, cloud, and cognitive computing, the provision of services such as access control and non-repudiation may help mitigate the vulnerability of computer systems despite the explosion in the expansion and dynamic state of the Internet, which enable the functioning of the systems. The fifth industrial revolution is the key to facilitating development, but its growth is faced with vulnerabilities to insecurity. While we can select and apply ways of combating insecurity, the factors that caused the vulnerability, such as interconnection of its components and overdependence on even its smallest components, have to be addressed separately. In addition, the means of mitigating them have to be determined, and well-developed strategies have to be put in place.

### 11.1. Old Technologies That Are Still Relevant to Cybersecurity

Despite the criticism and vulnerabilities in existing security technologies, systems such as GPS and biometric identification, which have been around for a while, have proven rather resilient but important in their respective fields. GPS systems that are now being adopted even by the automobile industry have been key in location determination and navigation. Through this, it can be used to determine the origin and destination of any other GPS enabled systems or to calculate routes to destination. These are important to security, as the location data of the physical security components is a cause of concern. Biometric devices have also been widely used, as they have helped reduce unauthorized access to vital components that can be used to store data.

### 11.2. New Security Promises That Have Been Fulfilled

With an increase in the number of IoT devices, there has been an increase in entry points for malicious activity. In response, smart cities are adopting IoT sensors in addition to existing systems. These sensors are connected to a broader network to enable aggregation and analysis of the collected data. Moreover, smart cities need to adopt network layers [124] with a central core from which all data and communication updates are processed. However, the components that are integrated within such a network would need to be scalable, interoperable, and compliant to ensure security maintenance. Blockchain, which is a recent invention, is proving to be viable in cybersecurity and in preventing a denial of service (DoS) with the help of the distributed ledger technology found in cryptocurrencies. Its decentralized nature provides data security and also reduces the reliance of a network on a central authority.

Users of cybersecurity systems have indicated that there is not a single defense against cybersecurity threats that can be completely relied upon, as what is considered safe at a particular moment may not be in a future date. This is due to the dynamic nature of cybercrime and the ingenuity of cybercriminals. Blockchain has been deemed to have the capabilities of data integrity and confidentiality; however, much of it is still being criticized with respect to its reliance.

### 11.3. Above New and Old Technologies

Finally, security and safety of information and data shared over a computer-based system and those of the person sharing them should be of the most significant concern. Therefore, cybersecurity must be seriously addressed by all concerned parties to develop a safe and secure environment in which smart people can thrive in a smart city.

## Figures and Tables

**Figure 1 sensors-23-00120-f001:**
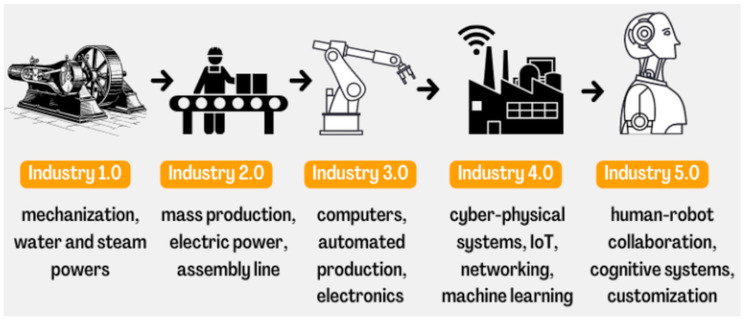
Industrial revolution [9].

**Figure 2 sensors-23-00120-f002:**
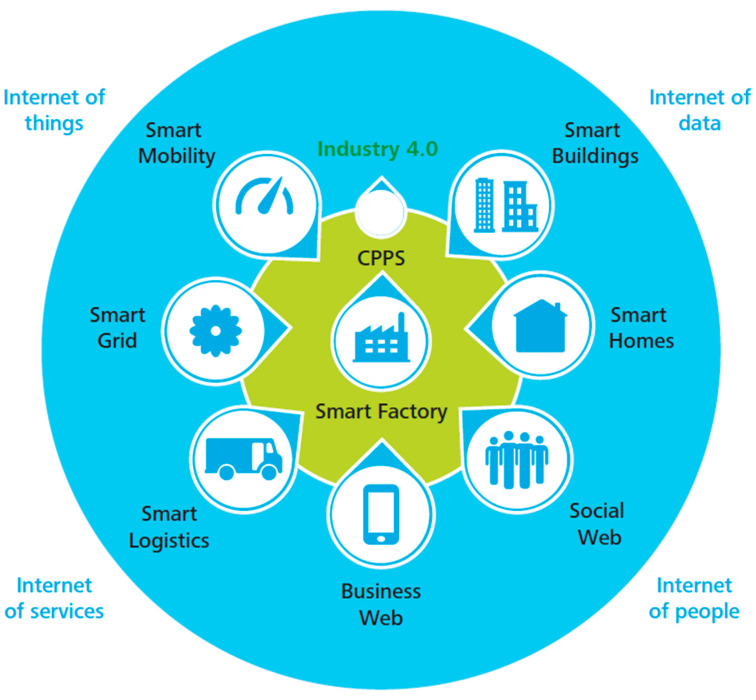
The Industry 4.0 Environment [26].

**Figure 3 sensors-23-00120-f003:**
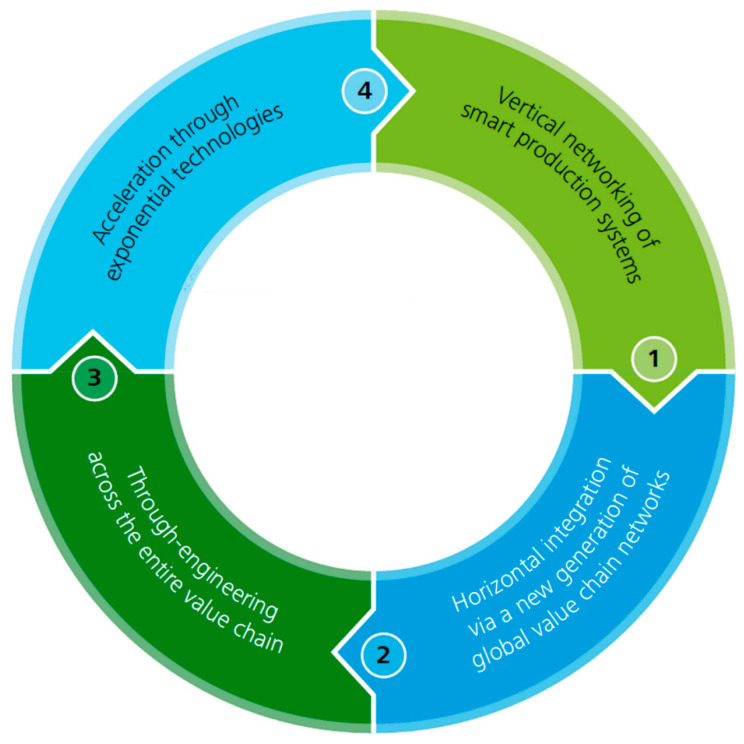
Elements of Industry 4.0 [26].

**Figure 4 sensors-23-00120-f004:**
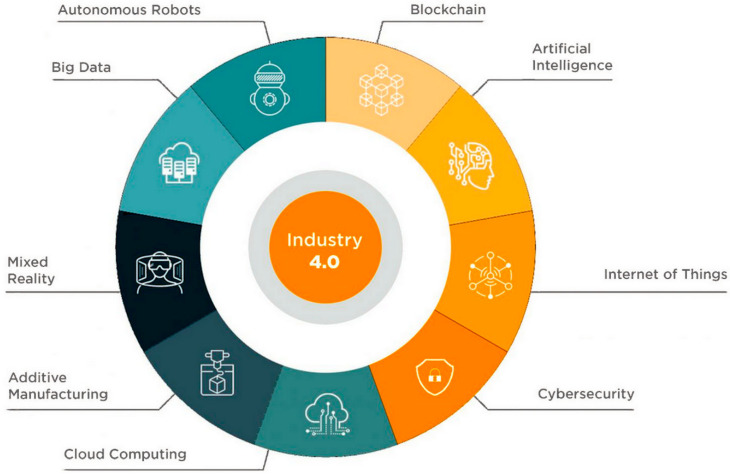
Industry 4.0 and the elements of digitalization [38].

**Figure 5 sensors-23-00120-f005:**
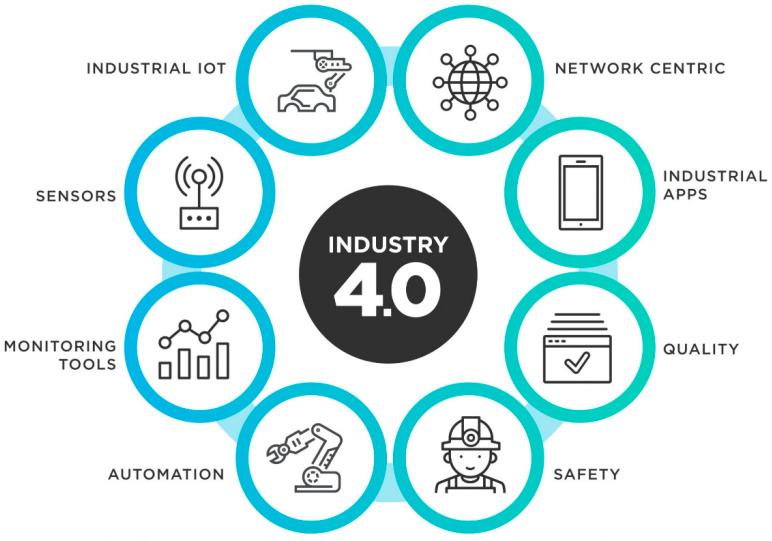
Tools of digitalization in industry 4.0 [45].

**Figure 6 sensors-23-00120-f006:**
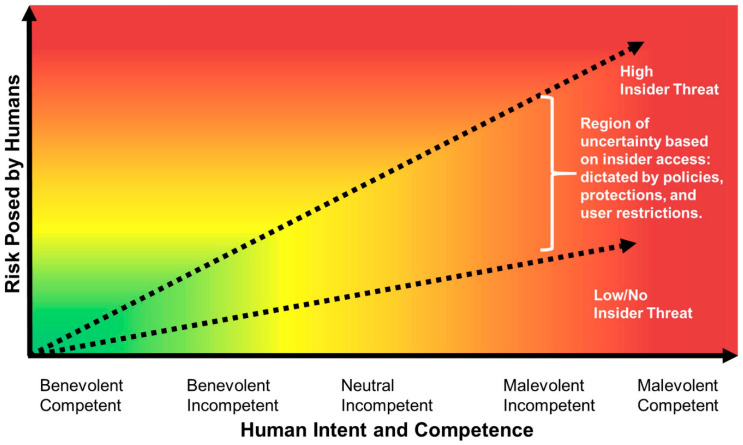
Representation of cyber security risks posed by humans considering the benign or malicious intent and the level of computational competence [57].

**Figure 7 sensors-23-00120-f007:**
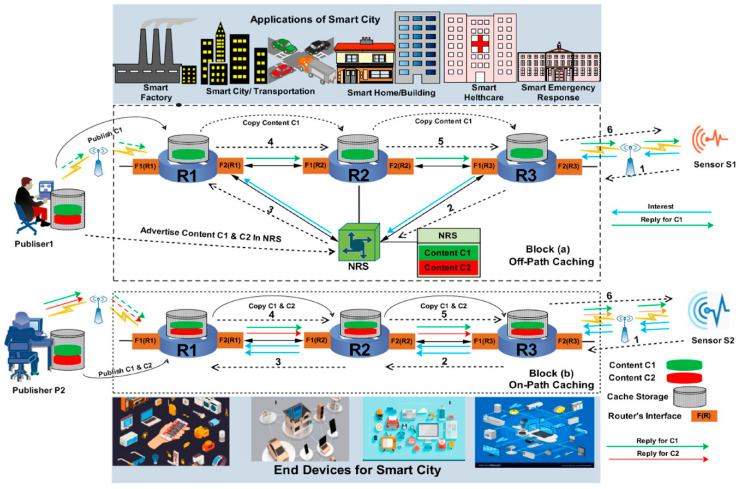
Representation of off-path and on-path caching mechanism in smart city [87].

**Figure 8 sensors-23-00120-f008:**
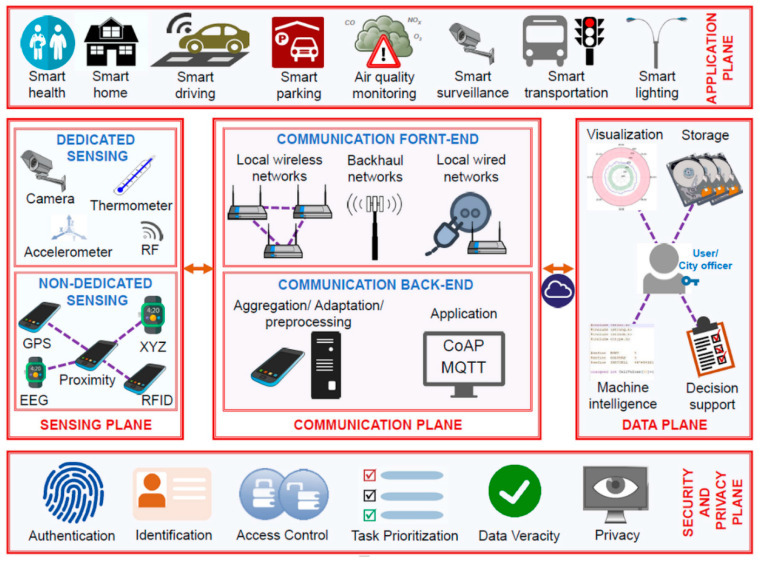
Architectural depiction of smart city [96].

**Table 1 sensors-23-00120-t001:** Key-words used for searching.

	Main Terms		Additional Terms		
Terms grouped	“cybersecurity” related	“smart city” related	actors involved related	technologies involved related	discussion/approaches related
Key-words	cyber risk, cybersecurity, cyber threats, cyber-attacks etc.	smart city, smart cities, smart grid, sustainable city etc.	cybercrime, psychology of cybercriminals etc.	MQTT, 6G, 5G, LoRaWAN, 6LoWPAN, RPL, blockchain etc.	IoT, Big Data, Artificial Intelligence, Machine Learning in cybersecurity, cobots, standards, classifications etc.

**Table 2 sensors-23-00120-t002:** Articles from indexed journals (per publisher).

	Items	Published Interval	Citations
Annual Review of Sociology	1	2012	160
Frontiers	5	2018	169
Hindawi	2	2018, 2022	243
IEEE	1	2020	524
Nature—Scientific Reports	1	2016	34
OMICS: A Journal of Integrative Biology	1	2018	310
The Science and Information (SAI) Organization	1	2016	102
Online-Journals.org	1	2017	1082
Sage	1	2017	52
Springer	2	2016, 2022	59
Taylor & Francis	2	2018, 2019	193
Sciencedirect	9	2014–2022	1089
MDPI	53	2017–2022	4413

**Table 3 sensors-23-00120-t003:** Types of items cited in the review.

	Published in (year)	No. of Items	Citations	Average (cit. per item)
Articles in scientific Journals	2012–2022	80	8480	106
Proceeding of conferences	2015–2021	14	1701	122
Books & chapters in books	2010–2022	8	617	77
Companies reports	2015–2022	12	463	39
Agencies reports	2016–2021	6	136	23
Blogs (from professionals)	2022	2	-	-
PhD Thesis	2018	1	6	6
Standards (ISO)	2019	1	-	-
TOTAL	2012–2022	124	11403	92

## Data Availability

Not applicable.

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
