# Peer review of "The Need for Cybersecurity in Industrial Revolution and Smart Cities"

_sensors, 2022, doi:10.3390/s23010120_

Round 1
Reviewer 1 Report
The article reviews in the context of Smart Cities platforms that aggregate data from sources such as sensors and other mobile devices that are connected to the Internet of Things to easily access data that can be used for decision making and thus solving these problems. The emphasis is on addressing security and privacy challenges.
The problem discussed is relevant, and the article is well written.
Observations:
The authors should detail in “2. Methodology” the review protocol (i.e., how they selected the articles): the eligibility criteria, the information sources (e.g., electronic databases), the search strategy, the selection process, etc.;
Papers from the past two years should be included.
Author Response
Dear Professor,
First of all, I have to thank you for your suggestions to improve the work.
Regarding your suggested improvements:
- Last 2 years papers have been included (most of them: added, very few: replaced and the text updated)
- Methodology part: few tables and words were added, as well as an Annexa 1 – document (with more information about references)
The improved article (next draft) has been uploaded.
Thank you very much for your suggestions, in hoping that all of your observations are, at least, enough addressed,
Yours,
authors
Reviewer 2 Report
* The concept and idea are good
* Some changes to be made in the abstract part, Abstract should have a clear intention of this paper. It seems to be confusing. Please rewrite the same
* In the Methodology section, the initial paragraphs seems to be unclear and needs a re-writing.
* The paper focuses in explaining industry 4.0 and 5.0 in depts. But not much thought process is presented on the smart cities. The same has to be addressed.
* The citations in the document is referring the papers published till 2020. IS there no work done in 2021 and 2022 (almost 2 years?) The recent proposals to be added and referred accordingly.
*Since it is a review kind of paper, the authors are requested to present the comparison study of the papers in a tabular format that makes the study findings visible.
Author Response
Dear Professor,
First of all, I have to thank you for your suggestions to improve the work.
Regarding your suggested improvements:
- The Abstract has been improved (a little bit) to be more clear
- Methodology part has been improved: the text has been corrected, few tables and words were added, as well as an Annexa 1 – document (with more metrics/information about references)
- About industry 4/5 vs smart cities, there is a long story. The shorten version of the story is that we had decided to split the initial work (actually the second major draft which had around 60 pages in 2021) in 4 parts: blockchain; technical short approaches of cybersecurity; smart cities focused- still in draft; (and) industry 4/5 focused in the context of smart cities – this article. Considering the vastity of this subject, some choices in order to reduce/concentrate the study were made with the costs of a little superficiality in approaching smart cities cybersecurity perspective.
- Last 2 years papers have been included (most of them: added, very few: replaced and the text updated): 9 references 2021 plus 12 from 2022 (as percent of 18% from total)
- Addressed in 2. : “comparison study of the papers in a tabular format”.
The improved article (next draft) has been uploaded.
Thank you very much for your suggestions, in hoping that all of your observations are, at least, enough addressed,
Yours,
Authors
Reviewer 3 Report
This paper review paper evaluates the aggregate platforms and other mobile devices connected to IoT. The paper is motivated by the fact that these platforms can be vulnerable to attacks and insecurity. Thus, they will need cybersecurity to overcome the security issues. After reading this paper, I have the following comments:
Major Comments:
- The abstract is not concise. You have to focus your abstract on the title of the paper.
- The paper is not well motivated to introduce such a review article.
- There must exist some motivational scenarios in the Introduction section.
- Evaluating the aggregate platforms is realized in this paper.
- Also, reviewing or comparing the published papers in the field of cybersecurity in industrial revaluation and smart cities are not realized in this paper.
- The figures introduced in this paper are already online and related to some well-known websites. The authors' contribution to the figures is missing.
- I have never seen a review paper with zero tables. There must exist some table that introduces comparisons for the previously published contributions in the field of review.
- Cybersecurity is a huge field. This review paper has not selected one direction for example access control and efficiently sorting it.
Minor Comments:
- There are some typos and grammatical issues in this paper.
The figures’ quality is low.
Author Response
Dear Professor,
First of all, I have to thank you for your suggestions to improve the work.
Regarding your suggested improvements:
A.Minor comments:
A.1. The English used has already been improved (I have to admit there I found around 400 mistakes – an average of 20 per pages).
A.2. Figures have been updated (few replaced) as relevance and quality.
- Major comments:
B.1. The Abstract has been improved (a little bit) to be more clear
B.2. In Introduction section we had modified the text a little, to be clearer as message/motivation.
B.3. The figures are, from our perspective, not the best but well suited to the subject. Adding/modifying them to be more suited to the scope, it took from us to many hours of “graphical” efforts for – in our opinion – a small effect. Of course, if you insist, I would make this, because seems to be a “nice to have” think.
B.4. Methodology part has been improved: the text has been corrected, few tables and words were added, as well as an Annexa 1 – document (with more metrics/information about references)
B.5. About industry 4/5 vs smart cities, there is a long story. The shorten version of the story is that we had decided to split the initial work (actually the second major draft which had around 60 pages in 2021) in 4 parts: blockchain; technical short approaches of cybersecurity; smart cities focused- still in draft; (and) industry 4/5 focused in the context of smart cities – this article. Considering the vastity of this subject, some choices in order to reduce/concentrate the study were made with the costs of a little superficiality in approaching smart cities cybersecurity perspective.
B.6. Last 2 years papers have been included (most of them: added, very few: replaced and the text updated): 9 references 2021 plus 12 from 2022 (as percent of 18% from total)
The improved article has been uploaded.
Thank you very much for your suggestions, in hoping that all of your observations are, at least, enough addressed,
Yours,
Authors